# Evidence for the weakly coupled electron mechanism in an Anderson-Blount polar metal

N.J. Laurita[1,2], A. Ron[1,2], Jun-Yi Shan [1,2], D. Puggioni[3], N.Z. Koocher[3], K. Yamaura [4], Y. Shi[5], J.M. Rondinelli [3] & D. Hsieh[1,2]

Over 50 years ago, Anderson and Blount proposed that ferroelectric-like structural phase transitions may occur in metals, despite the expected screening of the Coulomb interactions that often drive polar transitions. Recently, theoretical treatments have suggested that such transitions require the itinerant electrons be decoupled from the soft transverse optical phonons responsible for polar order. However, this decoupled electron mechanism (DEM) has yet to be experimentally observed. Here we utilize ultrafast spectroscopy to uncover evidence of the DEM in $LiOsO_3$, the first known band metal to undergo a thermally driven polar phase transition ($T_c \approx 140$ K). We demonstrate that intra-band photo-carriers relax by selectively coupling to only a subset of the phonon spectrum, leaving as much as 60% of the lattice heat capacity decoupled. This decoupled heat capacity is shown to be consistent with a previously undetected and partially displacive TO polar mode, indicating the DEM in $LiOsO_3$.

[1] Department of Physics, California Institute of Technology, Pasadena, CA 91125, USA. [2] Institute for Quantum Information and Matter, California Institute of Technology, Pasadena, CA 91125, USA. [3] Department of Materials Science and Engineering, Northwestern University, Evanston, IL 60208-3108, USA. [4] Research Center For Functional Materials, National Institute for Materials Science, 1-1 Namiki, Tsukuba, Ibaraka 305-0044, Japan. [5] Institute of Physics, Chinese Academy of Sciences, Beijing 100190, China. Correspondence and requests for materials should be addressed to D.H. (email: dhsieh@caltech.edu)

Ferroelectric transitions—in which a crystal spontaneously develops a switchable electric polarization—are typically driven by long-range Coulomb interactions, and are therefore conventionally found in insulating dielectrics where such fields are unimpeded[1]. In metals, these interactions are immediately screened by the itinerant electrons, seemingly suggesting an incompatibility between metallicity and polarity[2]. However, recent theoretical treatments[2–4] have suggested an alternative route by which metals may achieve polar order, where polar instabilities are instead driven by short-range interactions originating from the local bonding environment of the cations in the unit cell. An experimental signature of these so-called geometric polar metals is naturally encoded in their electron–phonon interactions, as the viability of this mechanism is believed to hinge on a decoupling[5] of the itinerant electrons from the displacive transverse optical (TO) polar phonons which drive the transition. This underlying concept was first noted in Anderson and Blount's seminal 1965 proposal[6], but recently recast by Puggioni and Rondinelli[4] as a guiding operational principle in the design of polar metals. Despite its fundamentality, this decoupled electron mechanism (DEM) has never been experimentally verified due to the scarcity of metals which display intrinsic polar transitions. Thus, the strength of itinerant electron–polar phonon interactions and the mechanism by which metals may undergo polar transitions is currently unresolved.

An excellent testbed for uncovering the nature of itinerant electron–polar phonon interactions, and by extension the DEM, is LiOsO$_3$. This material shares an identical crystal structure to LiNbO$_3$-type ferroelectrics[7], and exhibits an analogous polar transition from the centrosymmetric $R\bar{3}c$ (Fig. 1a) to polar $R3c$ (Fig. 1b) space groups driven primarily by Li ion displacement along the trigonal [001] polar axis[8,9]. However, unlike LiNbO$_3$, LiOsO$_3$ is metallic in both the nonpolar and polar phases[8]. Density functional theory calculations (Fig. 1c) suggest that this metallicity derives from the presence of O $2p$ and possibly correlated[8–10] Os $5d$ $t_{2g}$ orbitals at the Fermi level (see Supplementary Note 1). The fact that the polar transition is driven by Li ion displacement while the metallicity derives from O and Os orbitals is suggestive that the DEM may occur in LiOsO$_3$. However, the expected $A_{2u}$ TO soft mode[11] associated with the Li ion displacements was not detected by Raman spectroscopy[12], leaving both its coupling to the itinerant electrons and the displacive versus order–disorder character of the transition debated[9,10,13,14].

Ultrafast optical pump–probe experiments are capable of ascertaining how efficiently photo-generated carriers relax via various phonon decay channels[15,16], and are therefore well-suited to study how the electron–phonon coupling strength varies across different phonon modes in LiOsO$_3$ (Fig. 1d, e). Here, we utilize this technique to uncover evidence of the DEM in LiOsO$_3$. In our experiment, a pump photon energy of 1.56 eV was chosen so as to only generate photo-excitations within the metallic band, presumably via dipole allowed Os $5d$–O $2p$ transitions (Fig. 1c), while still exceeding the maximum phonon energy of LiOsO$_3$[12] so as to avoid any restrictions on the photo-carrier - phonon scattering phase space. The phonon-mediated photo-carrier relaxation dynamics were then tracked via the pump induced fractional change in reflectivity ($\Delta R/R$), as measured by a time-delayed probe pulse of tunable energy, although the relaxation dynamics were not found to vary significantly within our accessible energy range (see Supplementary Note 2).

## Results

**Temperature dependence of the relaxation dynamics**. The measured temperature dependent reflectivity transients of LiOsO$_3$ are shown in Fig. 2a. At all temperatures, $\Delta R/R$ displays an abrupt drop at time $t = 0$ followed by a recovery on the picosecond time scale to a negatively offset value. Both the magnitude of the drop and the recovery dynamics are clearly sensitive to $T_c$. To better highlight the temperature dependent

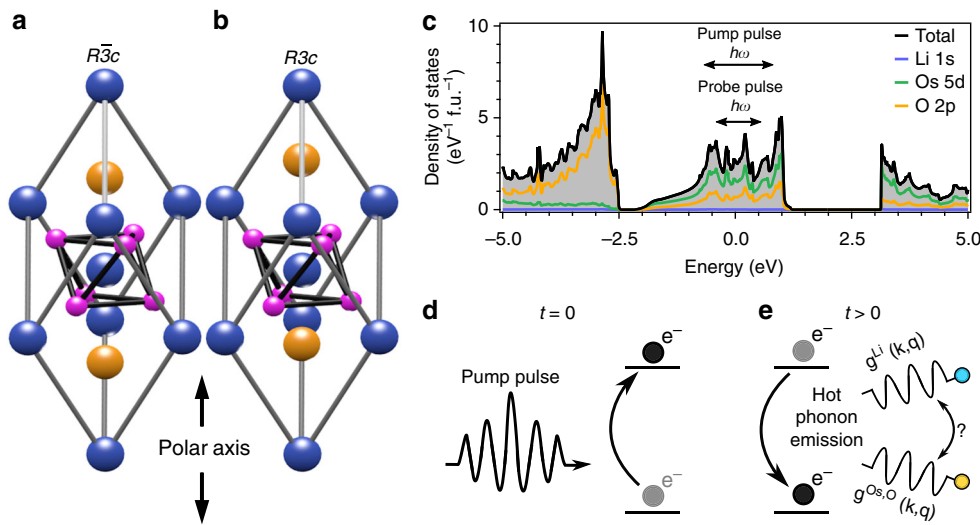

**Fig. 1** Scheme for probing electron–phonon interactions of LiOsO$_3$ Crystal structure of LiOsO$_3$ in **a** nonpolar $R\bar{3}c$ and **b** polar $R3c$ phases with blue, orange, and pink spheres representing Os, Li, and O atoms, respectively. These phases are distinguished by the displacement of the Li ions along the polar axis (black arrows). **c** Density functional theory calculation of the orbital resolved electronic density of states of the $R3c$ structure of LiOsO$_3$ (see Supplementary Note 1). Our pump (1.56 eV) and probe (0.92 eV) pulse energies permit excitation and monitoring of only intra-band photo-excitations, as shown by the black arrows in the inset. **d**, **e** A cartoon of our experimental scheme for probing itinerant electron–polar phonon interactions in LiOsO$_3$. **d** An intense pump pulse generates electronic excitations within the metallic band of LiOsO$_3$ at an initial time $t = 0$. **e** These excitations relax at later times $t > 0$ by coupling to the lattice, naturally embedding the strength of the electron–phonon coupling $g(\mathbf{k}, \mathbf{q})$ in their relaxation rate. By comparing the temperature dependent relaxation rate to that expected of polar phonons, we may identify which phonon modes, either the $A_{2u}$ polar mode (blue sphere) or O and Os modes (orange sphere), primarily mediate photo-carrier relaxation and therefore are most strongly coupled to the itinerant electrons of LiOsO$_3$

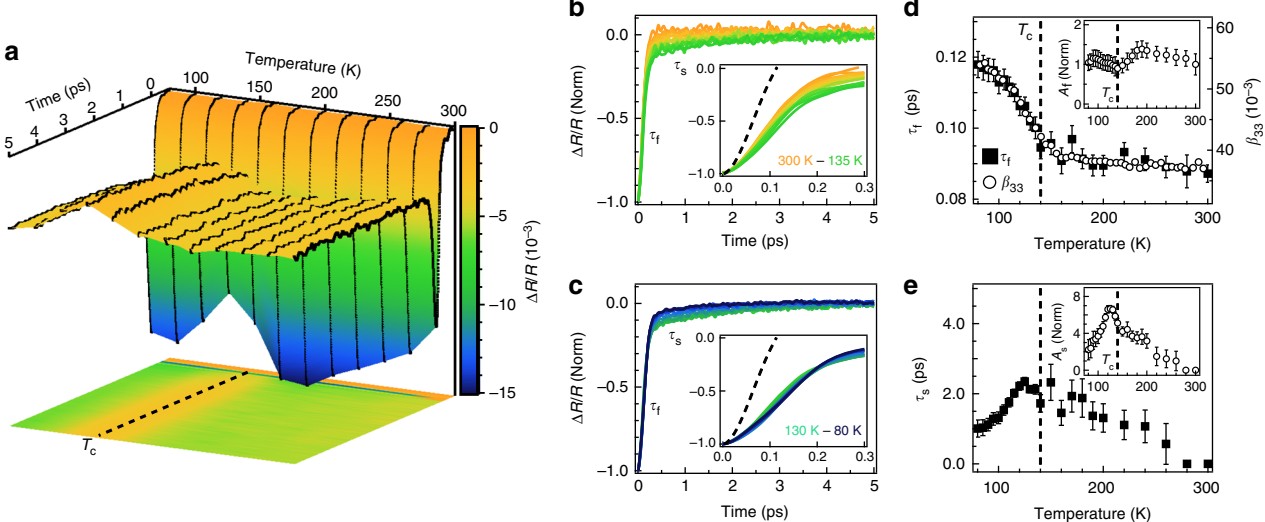

**Fig. 2** Temperature dependent relaxation dynamics of LiOsO$_3$. **a** Three-dimensional surface plot of the transient reflectivity $\Delta R/R$ of LiOsO$_3$ as a function of temperature and time delay. Black lines are raw traces at select temperatures. An image plot of this data is projected at the bottom where clear signatures of the polar transition are observed at $T_c \approx 140$ K. **b**, **c** Normalized reflectivity transients after background subtraction showing two distinct relaxation timescales, a faster relaxation with time constant $\tau_f$ and a slower relaxation with time constant $\tau_s$, in **b** nonpolar and **c** polar phases. Insets: magnified view of the fast decay process compared to the Gaussian instrument resolution of our experiment, shown as a black dashed line, demonstrating our experiment is not resolution limited (see Supplementary Note 3). **d**, **e** Results of modeling the reflectivity transients with a bi-exponential function. **d** Temperature dependence of $\tau_f$ plotted with the anisotropic displacement parameter $\beta_{33}$, a measure of the structural order parameter[8]. Inset: Temperature dependence of the fast relaxation amplitude $A_f$ normalized by its 300 K value. **e** Temperature dependence of $\tau_s$. Inset: temperature dependence of the slow relaxation amplitude $A_s$ normalized by its 260 K value, the temperature at which it is first discernable. Error bars in **d**, **e** derive from the $\chi^2$ of the bi-exponential fits

photo-carrier relaxation dynamics, we subtract the offset from each transient and normalize their magnitudes (Fig. 2b, c). For $T \gg T_c$, only a single fast relaxation process with a time constant of $\tau_f \approx 0.1$ ps is observed, a typical time scale for electronic relaxation in metals[16]. However, as $T_c$ is approached, an additional slow relaxation process with time constant $\tau_s \approx 1$–3 ps emerges and peaks in magnitude near the polar transition. As we demonstrate below, the emergence of this slower relaxation process is the result of a decoupled TO polar mode which displacively softens across $T_c$.

To capture the temperature dependence of these two relaxation processes, we fit the reflectivity transients to a phenomenological bi-exponential model given by $\Delta R/R = A_f \exp(-t/\tau_f) + A_s \exp(-t/\tau_s) + C$, where $A_f$ and $A_s$ are the amplitudes of the fast and slow relaxation components, respectively and the constant offset $C$ accounts for slow ($\approx 20$ ns) heat diffusion out of the probed region of the sample (see Supplementary Notes 4 and 5). We begin by highlighting the temperature dependence of $\tau_f$ (Fig. 2d), which is found to closely track the structural order parameter as captured by the anisotropic thermal parameter $\beta_{33}$[8]. While this resemblance would appear to suggest that the relaxation dynamics are strongly tied to the polar transition, the order-parameter-like increase of $\tau_f$ is relatively modest ($\approx 25\%$). This is far weaker, for instance, than the $\approx 500\%$ divergence in photo-carrier lifetimes exhibited by the parent compounds of the pnictide superconductors at their structural transitions[17,18], and therefore not necessarily inconsistent with the DEM. Indeed, weak coupling between the electronic structure and the polar transition is further supported by the temperature dependence of $A_f$ (Fig. 2d inset), which is a measure of the change in the joint density of states at the probe wavelength, and therefore exceptionally sensitive to subtle variations in the electronic structure. Despite this sensitivity, $A_f$ is found to exhibit weak temperature dependence across $T_c$, exemplifying the insensitivity of the low-energy band structure to the polar transition, as

expected from first principles calculations[9,11] and consistent with the DEM[4,6].

In contrast to the fast relaxation component, the slow relaxation component displays stronger temperature dependence, as exemplified by both $A_s$ and $\tau_s$ which exhibit a cusp-like behavior across $T_c$ (Fig. 2e). However, as will be demonstrated below, this slow relaxation component does not result from electron–phonon interactions. This is in accordance with the general expectations of intra-band photo-carrier relaxation in metals, as the existence of multiple phonon decay channels in a hot metal does not result in separate relaxation processes, but instead, a single relaxation process with a relaxation rate given by a Matthiessen-type rule as $1/\tau_{tot} = 1/\tau_1 + 1/\tau_2 + \ldots$ etc. Instead, the emergence of a second relaxation component in metals is generally indicative of coupling to additional bosonic modes (e.g., magnons[19]), a gap induced inter-band relaxation bottleneck[20], or a preferred electron–phonon coupling in which only a subset of the phonon spectrum mediates photo-carrier relaxation[21].

**Microscopic origin of the relaxation dynamics.** To rule out a gap induced inter-band relaxation bottleneck, we performed measurements as a function of pump fluence, which are capable of distinguishing between single particle and bi-molecular relaxation processes[22,23]. Figure 3 displays the fluence dependence of the parameters of the bi-exponential model (see Supplementary Note 6). The amplitudes $A_f$ and $A_s$ (Fig. 3a, b) both exhibit the linear fluence dependence expected of the linear response regime. However, both $\tau_f$ and $\tau_s$ (Fig. 3c, d) are fluence independent within our error bars, which is inconsistent with the linear fluence dependence expected from bi-molecular recombination dynamics across a gap[23], and thus rules out a gap induced inter-band relaxation bottleneck as the origin of the slow relaxation process. With no other collective excitations aside from phonons known in LiOsO$_3$[8], we conclude that a preferred

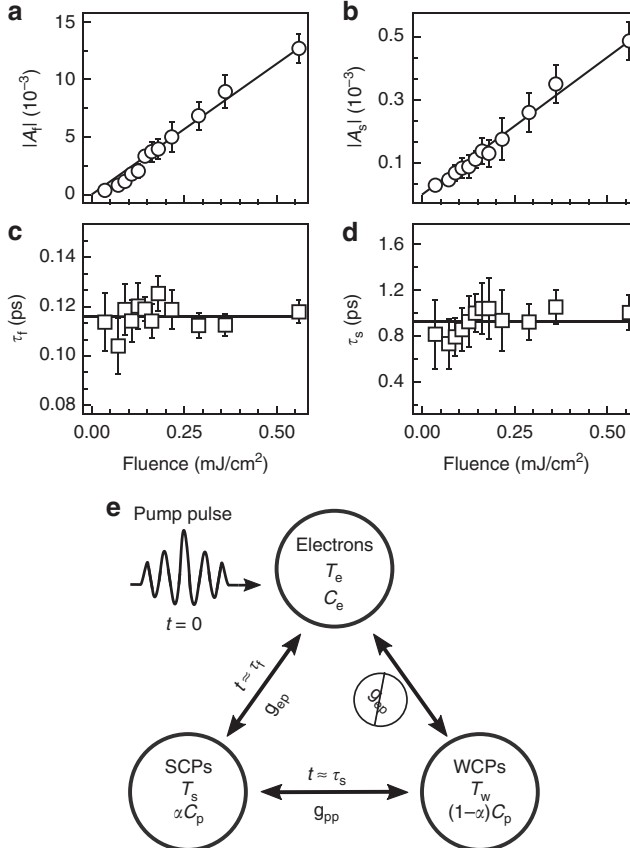

**Fig. 3** Microscopic origin of the two relaxations in LiOsO$_3$. **a–d** Fluence dependence of the transient reflectivity of LiOsO$_3$ at $T = 80$ K as captured by the parameters of the bi-exponential model. Both the amplitudes of **a** fast and **b** slow relaxation processes display a linear dependence that is characteristic of the linear response regime. However, the decay constants of **a** fast and **b** slow relaxation processes display a lack of fluence dependence that is indicative of intra-band photo-carrier relaxation and therefore suggests a selective electron–phonon coupling in LiOsO$_3$. Black lines in **a-d** are linear fits of the data while error bars derive from the $\chi^2$ of the bi-exponential fits. **e** Schematic of microscopic origin of the relaxation processes of LiOsO$_3$. Electrons, initially excited to a high effective temperature $T_e$ by the pump pulse, relax on the time scale $\tau_f$ by thermalizing with only a set of strongly coupled phonons (SCPs) at temperature $T_s$ via electron–phonon coupling $g_{ep}$. The SCPs then relax on a time scale $\tau_s$ by thermalizing with the rest of the lattice modes, referred to as the weakly coupled phonons (WCPs) at temperature $T_w$, via phonon–phonon coupling $g_{pp}$. The relative heat capacities of the two phononic thermal baths are found by partitioning the total lattice heat capacity $C_p$ by the parameter $\alpha$

electron–phonon coupling exists, in which photo-carriers selectively couple to only a subset of the overall phonon spectrum—referred to here as the strongly coupled phonons (SCPs)—before thermalization with the rest of the phonon modes—the weakly coupled phonons (WCPs)—occurs.

To identify which phonon modes strongly and weakly couple to excited photo-carriers, we appeal to a three-temperature thermalization model (TTM), which captures the selective electron–phonon coupling dynamics by treating the electrons, SCPs, and WCPs as three coupled thermal baths (Fig. 3e)[21]. In the TTM, excited photo-carriers, which have rapidly thermalized to a high electronic temperature (see Supplementary Note 7), relax by thermalizing with the SCPs via electron–phonon coupling $g_{ep}$, resulting in a single relaxation process with time

constant $\tau_f$. These SCPs then thermalize with the WCPs via anharmonic phonon coupling $g_{pp}$, prompting an additional relaxation process with time constant $\tau_s$. The temperature dependent heat capacities of the two lattice thermal baths are constructed by partitioning the reported[8] total lattice heat capacity $C_p$ by the parameter $\alpha < 1$, such that the SCPs carry heat capacity $C_s = \alpha C_p$, while the WCPs carry heat capacity $C_w = (1 - \alpha)C_p$. By solving the TTM under the experimentally defined initial conditions, we obtain the transient electronic ($T_e$), SCP ($T_s$), and WCP ($T_w$) temperatures (Fig. 4a), which are then combined via a conventional[16,21] weighted sum to form model reflectivity transients as

$$\frac{\Delta R}{R} = aT_e + b[\alpha T_s + (1 - \alpha)T_w], \tag{1}$$

where $a$ and $b$ are determined by the initial and final values of the experimental data. These model reflectivity transients are then fit to the experimental data (Fig. 4b), allowing for unique extraction of $g_{ep}$, $g_{pp}$, and $\alpha$ as fitting parameters (see Supplementary Note 8).

**Identification of the strongly and weakly coupled phonons.** With the thermalization model applied, we may now determine which phonon modes constitute the SCPs and WCPs. We begin by identifying the SCPs via a comparison of the extracted electron–phonon coupling function $g_{ep}$ to the phonon linewidths $\Gamma_{ph}$ of LiOsO$_3$. In first principles electron–phonon coupling theory[5], the strength of the coupling between the electronic structure and a particular phonon mode is naturally encoded in the phonon's lifetime. In the zero momentum limit, this manifests as a $g_{ep} \propto \sqrt{\hbar\Gamma_{ph}}$ scaling relation, thereby providing a route to identifying which modes primarily mediate photo-carrier relaxation. Among the phonons reported by Raman spectroscopy[12], the $^1E_g$ and $^2E_g$ modes exhibit a temperature dependent linewidth that appear to obey this scaling relation (Fig. 4c) (see Supplementary Note 9). This suggests that these modes, and possible others whose linewidths have not yet been reported, constitute the SCPs. We can ascertain the coupling strength to these modes by converting $g_{ep}$ into the dimensionless form $\lambda$ via the relation $g_{ep} = (6\hbar\gamma/\pi k_B)\lambda\omega^2$, where $\gamma$ is the Sommerfeld coefficient and $\omega^2$ is the second moment of the $^1E_g$ and $^2E_g$ phonon frequencies[15]. Through this analysis, we find a dimensionless coupling of $\lambda = 0.09$ at the polar transition of LiOsO$_3$, a value comparable to that of more conventional metals[24]. It should be noted that the $^1E_g$ and $^2E_g$ modes are primarily associated with distortions of the OsO$_6$ octahedra[12] (Fig. 4c inset) and are thus not associated with Li ion motion along the polar axis, consistent with the DEM.

To identify which modes constitute the WCPs, we examine the temperature dependence of $1 - \alpha$ (Fig. 4d), which represents the heat capacity of these weakly coupled phonons. Before proceeding, it should be noted that the presence of a selective electron–phonon coupling in LiOsO$_3$, i.e., an $\alpha < 1$, is in itself peculiar. Previously, such selective coupling has only been observed in materials such as graphite[21], iron pnictides[22], and cuprates[25], and has been attributed to their reduced effective dimensionality[26]. Essentially, their layered structures naturally give rise to a preferred coupling to in-plane rather than inter-plane phonon modes, resulting in a spatially anisotropic electron–phonon coupling. However, LiOsO$_3$ does not possess a layered structure and there has thus far been no evidence that it behaves as an effective 2D system[9], suggesting a distinct explanation for the observed selective electron–phonon coupling. Instead, the selective coupling is naturally explained by the DEM,

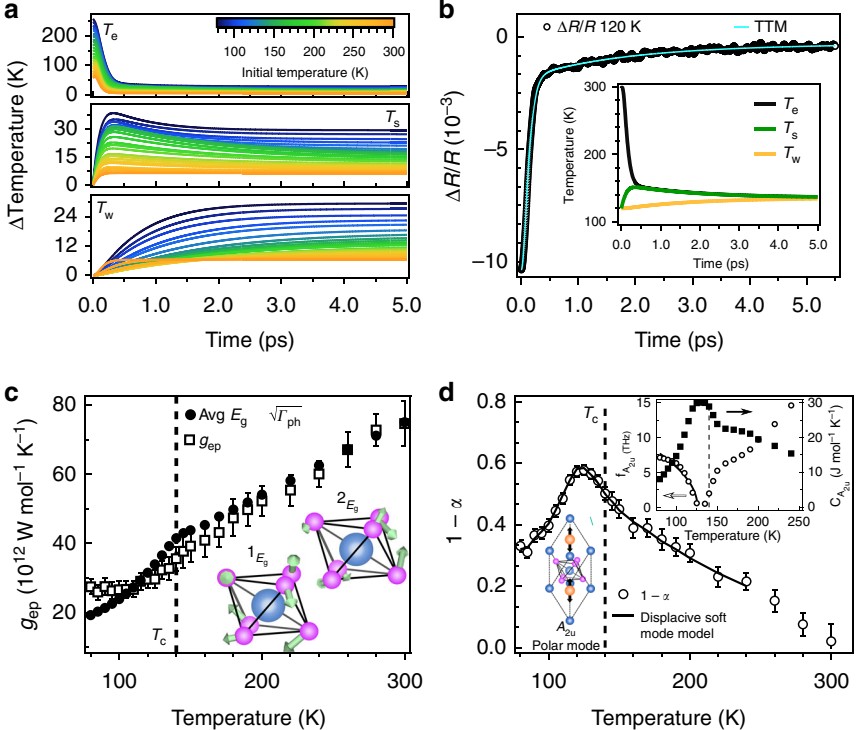

**Fig. 4** Identification of the strongly and weakly coupled phonons. **a** Transient changes in the electronic ($T_e$), strongly coupled phonon ($T_s$), and weakly coupled phonon ($T_w$) temperatures as extracted from a three-temperature thermalization model (TTM) of the relaxation dynamics of LiOsO$_3$. Color scale represents different initial experiment temperatures before the pump pulse arrives. **b** Representative fit of an experimental reflectivity transient at an initial temperature of $T = 120$ K (black circles) produced by the TTM (teal line). Inset: transient temperatures $T_e$, $T_s$, and $T_w$ from which the model reflectivity transient is generated. **c** Comparison of the extracted electron–phonon coupling function $g_{ep}$ with an average of the square roots of the $^1E_g$ and $^2E_g$ phonon linewidths $\Gamma_{ph}$ as reported by Raman spectroscopy[12], suggesting these modes are strongly coupled phonons. Inset: Real space distortions of these modes with blue and pink spheres representing Os and O atoms respectively. **d** Temperature dependence of the heat capacity of the weakly coupled phonons represented by $1 - \alpha$. The dashed vertical line denotes the reported value of $T_c$[8]. The solid black line is a fit to a model where this heat capacity is attributed to a displacive $A_{2u}$ polar mode (lower inset) that softens and hardens across the polar transition. Upper inset: Resonant frequency $f_{A_{2u}}$ (open circles) and heat capacity $C_{A_{2u}}$ (closed squares) of the polar mode extracted from the displacive soft mode model. The solid black line is a fit of the resonant frequency in the polar phase to the Cochran relation[27] $hf \propto \sqrt{1 - T/T_c}$ expected of polar soft modes, in which $T_c$ was shifted to coincide with the peak of $1 - \alpha$ at $T_c = 125$ K. All error bars derive from the $\chi^2$ of the three-temperature model fits

in which a weakly coupled polar mode first softens and then hardens across a partially displacive-like polar transition.

To demonstrate this, we show that the temperature dependence of $1 - \alpha$ is accounted for by the heat capacity of a displacive polar mode. We restrict this discussion to temperatures $T < 250$ K, below which the nonpolar optical modes are expected to be frozen out[12]. Therefore, in this regime the polar mode is the only thermally populated optical mode and we can approximate $1 - \alpha \approx C_{A_{2u}}(T)/C_p(T)$, where $C_{A_{2u}}(T)$ is the heat capacity of the $A_{2u}$ polar mode. We can then model $1 - \alpha$ by treating the polar mode as an Einstein phonon whose temperature dependent frequency $f_{A_{2u}}$ is then the only free parameter. Despite the simplicity of this model, we find that it not only completely reproduces the temperature dependence of $1 - \alpha$ (black line in Fig. 4d), but also allows for the extraction of the temperature dependent heat capacity and frequency of the polar mode (Fig. 4d inset), which clearly shows the cusp-like behavior across $T_c$ emblematic of displacive polar phonons (see Supplementary Note 10). The validity of this interpretation is further supported by the fact that the functional dependence of the extracted polar mode frequency below $T_c$ is well captured by the Cochran relation[27] $hf \propto \sqrt{1 - T/T_c}$ expected of polar soft modes. Furthermore, the extracted polar mode frequency of $f_{A_{2u}} \approx 7$ THz at our lowest measured temperature is in excellent agreement with zero temperature calculations, which predict a

polar mode frequency of $f_{A_{2u}} \approx 7.2$ THz (see Supplementary Note 11). This analysis not only demonstrates the partially displacive character of the transition, as opposed to the strictly order–disorder mechanism proposed due to the absence of a Li soft mode in the Raman spectra[12], but also shows that the photo-carriers couple extremely weakly to the polar mode (i.e., the polar mode belongs to the set of WCPs), thus indicating the DEM in LiOsO$_3$.

## Discussion

Having presented evidence for the DEM to occur in LiOsO$_3$, we now discuss the ramifications for polar materials in general. The DEM is contingent on a particular form of the electron–phonon coupling, in which the itinerant electrons are prevented from coupling to TO phonons by the polarization factor $\mathbf{q} \cdot \mathbf{e_q}$ in the electron–phonon coupling matrix elements, where $\mathbf{q}$ and $\mathbf{e_q}$ are the phonon wave vector and polarization, respectively[5]. Our results suggest this form of electron–phonon coupling to be an excellent approximation in LiOsO$_3$, and may be largely applicable to polar metals. However, while the itinerant electrons appear to be nearly decoupled from the polar transition, we cannot rule out a small but finite coupling that perhaps contributes to the modest 25% increase in $\tau_f$ in the polar phase (Fig. 2d). One way that such finite coupling may arise is from the longitudinal optical (LO)/TO degeneracy due to the screened Coulomb interactions in polar

metals, which was not accounted for in Anderson and Blount's original proposal[6], but is captured by Puggioni and Rondinelli's weak-coupling operational principles[4]. At finite carrier densities the LO/TO modes mix as $\mathbf{k} \to 0$, and thus a unique differentiation between the LO/TO modes participating in the loss of inversion symmetry is no longer possible (see Supplementary Note 11). Taken together, our experimental results support a picture in which polar transitions in metals are driven by short-range interactions[2,13] related to the bonding environment of the cations within the unit cell, which endure the metallicity by virtue of being decoupled from the electronic structure at the Fermi level.

## Methods

**Time-resolved reflectivity measurements.** Time-resolved reflectivity experiments were performed using a pump pulse with center wavelength 795 nm (1.56 eV) and duration $\approx 100$ fs produced by a regeneratively amplified Ti:sapphire laser system operating at a 100 kHz repetition rate. The probe pulse was produced by an optical parametric amplifier operating at the same repetition rate. By referencing a lock-in amplifier to the frequency that the pump beam is mechanically chopped (10 kHz), fractional changes in the reflectivity $\Delta R/R$ as small as $10^{-5}$ can be resolved. Temperature dependent measurements were performed with a pump pulse of fluence $F = 0.5$ mJ/cm$^2$, while the probe pulse had center wavelength 1350 nm (0.92 eV) and fluence $F = 10$ μJ/cm$^2$. Fluence dependent measurements were performed at $T = 80$ K by varying the pump pulse fluence while the probe pulse was maintained at center wavelength 1500 nm (0.83 eV) and fluence $F = 20$ μJ/cm$^2$. All pulses were focused at near normal incidence on the $[42\bar{1}]$ face of a single crystal sample of approximate dimensions 0.25 mm $\times$ 0.5 mm $\times$ 0.25 mm grown by a solid state reaction under pressure (see ref. 8 for details regarding sample preparation and characterization).

**Three-temperature model of the relaxation dynamics.** The three-temperature model assumes that excited photo-carriers thermalize with a set of strongly coupled phonons before thermalization with the rest of the lattice occurs. In the model, the total lattice heat capacity $C_\mathrm{p}$ is partitioned into two separate phononic thermal baths, such that the strongly and weakly coupled phonons carry heat capacities $C_\mathrm{s} = \alpha C_\mathrm{p}$ and $C_\mathrm{w} = (1 - \alpha)C_\mathrm{p}$, respectively, where the parameter $\alpha < 1$. In this fashion, $\alpha$ describes the portion of the total lattice heat capacity which participates in photo-carrier—lattice thermalization, and may thus be used to determine which modes couple strongly to the excited photo-carriers.

In the model, excited photo-carriers are assumed to immediately thermalize to a Fermi-Dirac distribution at a high electronic temperature given by[21]

$$T_{\mathrm{e,i}} = \frac{1}{\delta_\mathrm{s}} \int_0^{\delta_\mathrm{s}} \left[ \sqrt{T_\mathrm{i}^2 + \frac{2(1-R)F}{\delta_\mathrm{s}\gamma} \exp\left(-\frac{z}{\delta_\mathrm{s}}\right)} \right] dz, \tag{2}$$

where $T_\mathrm{i}$ is the temperature before pump excitation, $\gamma$ is the Sommerfeld coefficient, $R$ is the reflectivity at the pump wavelength, $F$ is the pump fluence, $z$ is the depth into the sample, and integration is performed over one penetration depth $\delta_\mathrm{s}$ at the pump wavelength. We estimate photo-carrier thermalization occurs within a few fs after pump excitation[28] (see Supplementary Note 7). Heat exchange between the electronic and two lattice thermal baths is then governed by the equations

$$2C_\mathrm{e}\frac{\partial T_\mathrm{e}}{\partial t} = -g_\mathrm{ep}(T_\mathrm{e} - T_\mathrm{s}) + I(t, z) + \nabla \cdot [\kappa_\mathrm{e}\nabla T_\mathrm{e}], \tag{3}$$

$$C_\mathrm{s}\frac{\partial T_\mathrm{s}}{\partial t} = g_\mathrm{ep}(T_\mathrm{e} - T_\mathrm{s}) - g_\mathrm{pp}(T_\mathrm{s} - T_\mathrm{w}), \tag{4}$$

$$C_\mathrm{w}\frac{\partial T_\mathrm{w}}{\partial t} = g_\mathrm{pp}(T_\mathrm{s} - T_\mathrm{w}), \tag{5}$$

where $C_\mathrm{e}$ is the electronic heat capacity, $I(z, t)$ is the laser source term, $\kappa_\mathrm{e}$ is the thermal conductivity, $g_\mathrm{ep}$ and $g_\mathrm{pp}$ are the electron–phonon and phonon–phonon coupling functions, and $T_\mathrm{s}$ and $T_\mathrm{w}$ are the temperatures of the strongly and weakly coupled phonons, respectively.

The three-temperature model equations are then solved to obtain the time dependent electronic and lattice temperatures. Model reflectivity transients are then constructed by convolving a normalized Gaussian with a conventional[16,21] weighted sum of the electronic and lattice temperatures as

$$\frac{\Delta R}{R} = aT_\mathrm{e} + b[\alpha T_\mathrm{s} + (1 - \alpha)T_\mathrm{w}], \tag{6}$$

where $a$ and $b$ are determined by initial and final values of the experimental reflectivity transients. The model reflectivity transients are then fit to the experimental data using a least squares regression algorithm with $g_\mathrm{ep}$, $g_\mathrm{pp}$, and $a$ as relaxed fitting coefficients (see Supplementary Note 8).

## Data availability

The datasets generated are/or analyzed during the current study are available from the corresponding author on reasonable request.

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

## Acknowledgements

This work was supported by the U.S. Department of Energy under Grant no. DE SC0010533. D.H. also acknowledges funding from the David and Lucile Packard Foundation and support for instrumentation from the Institute for Quantum Information and Matter, an NSF Physics Frontiers Center (PHY-1733907). N.J.L. acknowledges support from the Institute for Quantum Information and Matter Postdoctoral Fellowship. N.Z.K. was supported by U.S. DOE-BES Grant no. DE-SC0012375. D.P. and J.M.R. were supported by ARO (Award no. W911NF-15-1-0017). Y.G.S. was supported by the National Key Research and Development Program of China (Nos. 2017YFA0302901 and 2016YFA0300604). We thank Rohit Prasankumar for helpful conversations, Qingming Zhang for sharing unpublished Raman data, and Stefano Lupi for sharing unpublished reflectivity results.

## Author contributions

D.H. and N.J.L. conceived the experiment. N.J.L. and A.R. performed the time-resolved reflectivity measurements. J.S. determined the crystal alignment. D.P., N.Z.K. and J.M.R. performed the DFT calculations, discussed the data and soft mode theories, and commented on the manuscript. Y.S. and K.Y. prepared and characterized the sample. N.J.L. and D.H. analyzed the data. N.J.L. and D.H. wrote the manuscript.

## Additional information

**Competing interests:** The authors declare no competing interests.

