## [Peer Review File · Nature Communications]

Editorial Note: This manuscript has been previously reviewed at another journal that is not operating a transparent peer review scheme. This document only contains reviewer comments and rebuttal letters for versions considered at Nature Communications .

REVIEWERS' COMMENTS:

Reviewer #1 (Remarks to the Author):

The authors have carefully and thoroughly replied to all the points raised by myself and the other Referees. I find that the responses are generally adequate and sound. As far as the objections I raised are concerned, I think that the replies mostly answer to my criticism. In the following I reply to the different points

1) I am glad the authors could show some dependence on the probe energy of their spectra (even if in a limited range) which strengthens the claims of the original manuscript.

2) I can accept the compromise proposed by the authors in the discussion of the debate about the role of electronic correlations in LiOsO_3 . I agree that the inclusion of the electronic correlations and spin-orbit coupling in the theoretical analysis is not necessary as long as the three-temperature models can be applied and can provide information about the target of the paper.

I acknowledge the difficulties in obtaining experimental information about the Drude response.

3) I am fine with addition to the supplementary information.

4) I agree that in previous cases the separation between strongly coupled and weakly coupled phonons has been attributed to two dimensionality. However I am not completely sure that this interpretation is exhaustive and that the results presented are really a totally different story. However, I do not believe that this is a serious limitation for this manuscript. I still believe that some words to contrast previous claims with the present result would be helpful, also to avoid readers to underestimate the present results. I would also quote the review paper I mentioned (which the authors also quote in the reply to another Referee).

I am therefore positively impressed by the work done by the Authors to improve the manuscript and I am willing to suggest publication of the manuscript in Nature Communications. However I would suggest the authors two points

a) Even if the replies to my points (and to the other Referees) are mostly convincing, I find that some of the information contained in the replies is still lacking in the manuscript and I would suggest the authors to improve in this direction. I believe that some points can be useful to make the manuscript more convincing for a general audience.

b) I think it would benefit the reader to connect the original proposal by Anderson and Blount with that of Rondinelli and Puggioni. From the present manuscript one only reads that Anderson and Blount proposed a possibility, but it is not clear how this connects with the more practical proposal of Ref. 3. I apologize I did not make this point in my original report

Reviewer #3 (Remarks to the Author):

The authors have revised the manuscript taking account the comments from all the referees. This has appreciably improved the manuscript. Certain interpretations of the analysis have also been altered, such as disorder is no longer considered central to the temperature dependence of the phonon coupling and a more robust analysis of the specific modes is given.

The manuscript now presents a more clear advance in understanding polar metals.

Reviewer #4 (Remarks to the Author):

The authors have responded in detail to each of the comments of the reviewers. As a result the paper is much improved. I have no objection to publication in Nat. Comm.

Reply to Referee #1:

The authors have carefully and thoroughly replied to all the points raised by myself and the other Referees. I find that the responses are generally adequate and sound. As far as the objections I raised are concerned, I think that the replies mostly answer to my criticism. In the following I reply to the different points

1) I am glad the authors could show some dependence on the probe energy of their spectra (even if in a limited range) which strengthens the claims of the original manuscript.

2) I can accept the compromise proposed by the authors in the discussion of the debate about the role of electronic correlations in LiOsO_3 . I agree that the inclusion of the electronic correlations and spin-orbit coupling in the theoretical analysis is not necessary as long as the three-temperature models can be applied and can provide information about the target of the paper.

I acknowledge the difficulties in obtaining experimental information about the Drude response.

3) I am fine with addition to the supplementary information.

4) I agree that in previous cases the separation between strongly coupled and weakly coupled phonons has been attributed to two dimensionality. However I am not completely sure that this interpretation is exhaustive and that the results presented are really a totally different story. However, I do not believe that this is a serious limitation for this manuscript. I still believe that some words to contrast previous claims with the present result would be helpful, also to avoid readers to underestimate the present results. I would also quote the review paper I mentioned (which the authors also quote in the reply to another Referee).

We agree with the referee and have expanded the discussion on this point in our most recent version of our manuscript, see the paragraph beginning on line 155. We have also cited the reference (Ref. #26) suggested by the referee as part of this discussion. We thank the referee for this suggestion.

I am therefore positively impressed by the work done by the Authors to improve the manuscript and I am willing to suggest publication of the manuscript in Nature Communications. However I would suggest the authors two points

a) Even if the replies to my points (and to the other Referees) are mostly convincing, I find that some of the information contained in the replies is still lacking in the manuscript and I would suggest the authors to improve in this direction. I believe that some points can be useful to make the manuscript more convincing for a general audience.

We have now added excerpts from the replies where appropriate to reinforce the main text in the most recent version of our manuscript as suggested by the referee.

b) I think it would benefit the reader to connect the original proposal by Anderson and Blount

with that of Rondinelli and Puggioni. From the present manuscript one only reads that Anderson and Blount proposed a possibility, but it is not clear how this connect with the more practical proposal of Ref. 3. I apologize I did not make this point in my original report.

We agree with the referee. We now explicitly connect Anderson's original proposal to the more recent work of Puggioni *et al.* in a new introductory paragraph of our manuscript. We thank the referee for this suggestion.

Reply to Referee #3:

1) The authors have revised the manuscript taking account the comments from all the referees. This has appreciably improved the manuscript. Certain interpretations of the analysis have also been altered, such as disorder is no longer considered central to the temperature dependence of the phonon coupling and a more robust analysis of the specific modes is given.

The manuscript now presents a more clear advance in understanding polar metals.

We are happy to hear that the referee found the most recent version of our manuscript to present a clear advance in the understanding of polar metals. We thank the referee for their comments on our manuscript.

Reply to Referee #4:

1) The authors have responded in detail to each of the comments of the reviewers. As a result the paper is much improved. I have no objection to publication in Nat. Comm.

We are happy to hear that the referee found our manuscript to be much improved. We thank the referee for considering our manuscript.